# Numerical Simulation of Airflow Organization in Vulcanization Tanks for Waste Tires

**DOI:** 10.3390/polym17020232

**Published:** 2025-01-17

**Authors:** Tianxi Su, Yongzhi Ma, Baolin Wang, Xiaowen Luan, Hui Li, Xuelong Zhang

**Affiliations:** 1College of Mechanical and Electrical Engineering, Qingdao University, Qingdao 266071, China; sutianxi@qdu.edu.cn (T.S.); zhangxuelong@qdu.edu.cn (X.Z.); 2Qingdao Wanfang Recycling Technology Co., Ltd., Qingdao 266071, China; wangbl@qdecolan.com (B.W.); luanxw@qdecolan.com (X.L.); lih@qdecolan.com (H.L.)

**Keywords:** waste tire retreading, vulcanization tank, airflow organization, orifice plate, numerical simulation

## Abstract

Currently, in the domestic practice of retreading tires using vulcanization tanks, some tanks exhibit uneven temperature distributions leading to low retreading success rates. To address that, this paper simulated the temperature and velocity fields during the heating process of vulcanization tanks for waste tire retreading. The results indicated that a higher heating power reduces the time required for the vulcanizing agent to reach the vulcanization condition, but it also increases the difference in tire temperature in the tank, with a severely uneven distribution of the temperature field. Subsequently, to improve the uniformity of temperature distribution and enhance the retreading rate of waste tires, this paper proposed two types of orifice plates to adjust the airflow organization. The results show that both the plain orifice plate and the frustum cone orifice plate can enhance the uniformity of the temperature field within the vulcanization tank and reduce the temperature difference between tires. Moreover, at the same heating power, the presence of the orifice plates increases the rate of temperature increase in the tires and the vulcanizing agent compared to the original vulcanization tank, improving the thermal efficiency of the vulcanization tank heater.

## 1. Introduction

With rapid development and global industrialization, the growing world population’s demand for automobile production is increasing [1], and the disposal of waste tires has become a global environmental protection challenge. The stable growth of the automotive industry inevitably leads to a significant increase in the number of end-of-life tires (ELT) [2], most of which come from waste passenger cars and truck tires [3,4]. Waste tires, known as “black pollution”, pose significant challenges in recycling and disposal technology. According to statistics, over the next thirty years, the global waste output is expected to increase by 70%, reaching 3.4 billion tons [1]. Thanks to the continuously growing automotive industry, China alone discards a staggering 16 million tons of waste tires annually, while the global volume has reached an astonishing 1.5 billion tons [5,6,7]. Against this backdrop, the recycling of waste tires is particularly important [8].

The treatment of waste tires mainly includes renovation and reuse, that is, the utilization of reclaimed rubber from waste tires, pyrolysis, and direct incineration. The reclaimed rubber from waste tires uses physical means to crush, separate, and sort waste car tires, allowing the production of rubber particles and rubber powder with minimal pollution at room temperature. Although the production of reclaimed rubber powder from waste car tires can achieve the harmless treatment of this “black pollution”, it ultimately results in additives and degradation materials derived from the waste tire rubber that may leach into the environment [9].

Pyrolysis involves the controlled combustion of rubber waste without oxygen, leading to the decomposition of rubber into smaller components such as fuel oil, gas, carbon black, sulfur, and metals [10,11]. The main objective of pyrolysis is to extract energy from waste tire rubber. The fuel produced through this process is purified to remove sulfur, char, and ash, ensuring the production of high-quality fuel, suitable for enhancing engine performance. The gas produced during the pyrolysis process is used as for heat and electricity generation in power plants [12]. Additionally, the carbon black produced by pyrolysis can be mixed with plastics or EVA foam (Ethylene-Vinyl Acetate copolymer foam), or it can be further processed into activated carbon [13]. In summary, pyrolysis products have a wide range of applications with considerable utilization value, aligning with the trends of the times, saving energy, being environmentally friendly, and being one of the best ways to create a sustainable development-oriented society. However, this solution for waste tire management requires large-scale pyrolysis equipment, with high construction and operation costs (high temperature and low pressure), limiting its large-scale industrial application [14].

Incineration is a self-sustaining exothermic process that occurs above 400 °C, and since the calorific value of waste tires is higher than that of coal (18.6–27.9 MJ/kg), they are used for energy recovery. The calorific value of waste tires is 32.6 MJ/kg, which can be used as a fuel source for producing steam, electricity, pulp, paper, lime, and steel. Additionally, Oriaku et al. [15] reported the recovery of carbon black (CB) through the combustion of tires in limited air supply through incineration. The recovered material can be used in small-scale industries for the production of printing inks and paints. The main advantages of incineration are the low cost of energy production and the maximum recovery of heat. However, the atmospheric pollution caused by emissions of flue gas and particulate matter is a serious air pollution source that needs to be addressed [16].

Tire retreading, as the primary and most effective method used for recycling, has characteristics such as multiple retreading, low material consumption, low cost, and long service life. Retreading is the process of replacing the worn tread of discarded tires with new tread so that the tires can be reused [17,18]. In the tire retreading process, the first step involves a thorough inspection of the discarded tire body to assess its suitability for reuse. After this evaluation, the tire crown is separated from the tire body through grinding and subsequent repair. After the necessary repairs, a cushion rubber sheet is applied, and a pre-vulcanizing agent is used to vulcanize the tread rubber. As shown in Figure 1, the process involves applying a vulcanizing agent to the outer tire and then attaching it to the inner tire. Among them, vulcanizing agents play a role in significantly improving the physical and chemical properties of rubber, such as elasticity, strength, heat resistance, medium resistance, and durability, even if rubber is transformed from thermoplastic rubber to thermosetting rubber through vulcanization reactions, which play a crucial role in enhancing the adhesion, physical properties, and production efficiency of refurbished tires, while reducing energy consumption and costs. The final stage includes the vulcanization of the tread rubber and the final inspection of the product’s quality before it is ready for the market [19]. Studies have shown that each retreaded tire requires only 30% of the energy and 25% of the raw materials needed to produce a new tire [20,21]. Each retreading can regain 60–90% of the service life of a new tire, with an average driving mileage of 50,000–70,000 km. In 2019, the United States produced over 4.05 million metric tons of waste tires, while the European Union produced 3.56 million metric tons. In recent years, in the United States, given the increasing number of waste tires, more tires are being recycled or used as energy [22]. Currently, in China, there is a large output of waste tires [23], a low volume of retreading with a retreading rate of about 4%, and generally low retreading rates [24], which are somewhat behind those of developed countries; however, significant progress has been made compared to the past. In China, with the rapid growth in the number of vehicles, the generation of waste tires is also increasing at a double-digit rate, posing severe challenges for environmental protection and resource conservation. Therefore, promoting the circular utilization of waste tires not only helps alleviate the issue of rubber resource shortage in China but also reduces environmental pollution, promoting the construction of a circular economy and a conservation-oriented society.

There are mainly two methods of retreading, hot vulcanization and cold vulcanization. Hot retreading of waste tires refers to the traditional retreading method, with vulcanization temperatures generally around 145~155 °C, with the hot retreading process temperature being far above the critical temperature, causing significant damage to the tire body. The use of rigid molds in the hot retreading process can easily lead to tire deformation, thus generating internal stress, causing layering, shoulder voids, and a high probability of tire blowouts, affecting the service life of the tire which can only be used for passenger car tires [17]. In this method, a new rubber layer is formed on the tire, and the entire tire is vulcanized at 150 °C to 180 °C to mold the tread pattern. Hot vulcanization uses mature technology, with a long history, cheap equipment costs, lower investment costs. Some tires, like airplane tires, can only be retreaded with hot vulcanization. The cold vulcanization method, also known as the pre-vulcanization method, has a general vulcanization temperature below 120 °C and is suitable for commercial vehicle tires. Since the critical temperature for the denaturation of tire rubber is 120 °C, this means that above 120 °C, the physical performance indicators of the tire rubber significantly decrease. For example, the adhesion strength between the rubber and the framework material decrease in such cases, leading to tire body layer separation, voids, or even blowouts. Therefore, this method, compared to traditional hot retreading, does not damage the retreaded tire body and does not affect the service life of the tire body. The optimal vulcanization temperature for cold retreading is generally around 100 °C. It uses pre-molded vulcanized strips or rings that are applied to the polished old tire body, which is vulcanized at low temperatures in the cylinder, potentially saving energy and reducing the aging phenomenon caused by secondary vulcanization, thus protecting tire body quality and extending service life. Under normal conditions (the wear resistance of the pre-vulcanized tread rubber reaches 80,000 km or more), the service life of hot retreaded tires should be 60~80% of that of new tires, while tires retreaded using the pre-vulcanization method can approach 100%. However, the current domestic cold retreading technology is not mature, and the general uneven temperature distribution in cold retreading vulcanization tanks leads to a low tire retreading rate.

To explore and solve the mentioned issues, this paper studies the temperature and velocity fields in the vulcanization tank and the development process of tire temperature within the tank through numerical simulations in ANSYS Fluent 2022R1. The investigation scope of tire temperature includes temperature differences, average tire temperature, and the temperature of vulcanizing agent which plays multiple roles in tire retreading, including restoring tire performance, providing adhesion, improving durability, maintaining tread shape, and promoting environmental protection. Subsequently, based on the simulation results, two types of orifice plate structures are proposed, the flat orifice plate and the frustum cone orifice plate, which significantly improve the uneven temperature distribution in the vulcanization tank’s temperature field, laying the foundation for the advancement of the tire retreading industry in China.

## 2. Materials and Methods

The vulcanization tank, as shown in Figure 2a,b, has two heating air ducts in the heating part of the tank body. The specifications of the screw-type armored thermocouple temperature meter are M27·1.5, which means the nominal diameter is 27 mm and the pitch is 1.5 mm. It is worth mentioning that the length of the thermometer is 150 mm. The temperature error is ±0.35 °C. The air flow direction and the position of the thermometer are shown in Figure 3. The schematic diagram of the energy transfer is shown in Figure 4. The heated air circulates clockwise, absorbs heat through an electric heating tank, and then releases heat through waste tires, repeating the cycle. Schematic diagrams and simplified geometric structures of the vulcanization tank are shown in Figure 2b,c. The detail can be seen in Table 1. The vulcanization tank consists of a tank, air ducts, and a driving fan, all made of 304 stainless steel [20,21]. The temperature of the laboratory environment is 7 °C. Each heating flue is equipped with three U-shaped heating tubes, the model of the waste tires are 1200r20 tires, and the operation procedure of the vulcanization tank includes the following steps:Close the vulcanization tank with 21 waste tires;Add compressed air at a pressure of 6 atmospheres to the tank;Turn on the driving fan and heating power to cause the air inside the tank to circulate and heat up; the heating power is 88 kw and the heating duration is 1600 s;Evenly heat the waste tires in the tank to facilitate the vulcanization of the vulcanizing agent.

## 3. Geometric Model

All experimental models are simulated through ANSYS Fluent 2022R1. To simplify the model, each U-shaped heating tube is modeled as two longitudinal heating tubes, ignoring the influence of the electronic control system. The relevant physical parameters of the material and the boundary conditions for simulation are shown in Table 2. It is worth noting that the power of the heat source is set as P = 45, 60, 75, 90 kW. Among them, the heating conditions are that the thermometer temperature < 100 °C and the tire temperature <80 °C. The driving fan operates with a flow rate of 4500 m^3^/h, with the minimum Reynolds number being 17,663, which is far greater than 4000, thus, the model is a turbulent flow model. Additionally, there are two other types of airflow organization orifice plate structures, which are the flat orifice plate and the frustum cone orifice plate, as shown in Figure 5 and Figure 6. The 5 cm aperture diameter and the 2 mm thickness of the orifice plate can be ignored. The diameter of the flat orifice plate is 1460 mm. The bottom diameter of the frustum cone orifice plate is 560 mm, and the diameter of the upper circle is 230 mm.

From left to right, there are 28 temperature monitoring points set for the vulcanizing agent, with their specific locations shown in Figure 7.

A simulation model of the waste tire curing tank is presented in the text, which includes fluid flow physical problems and fluid–solid heat transfer physical problems. In the computational domain, air is assumed to be incompressible and with constant properties. The flow is considered to be three-dimensional and steady. This study applies three governing equations.

The continuity equation and turbulence momentum equation are expressed as follows:(1)$$\rho \frac{\partial u_{i}}{\partial x_{i}}=0$$(2)$$\rho \frac{\partial \left(u_{i}u_{j}\right)}{\partial x_{i}}=-\frac{\partial P}{\partial x_{j}}+\frac{\partial }{\partial x_{i}}\left(u\left(\frac{\partial U_{i}}{\partial x_{j}}+\frac{\partial u_{j}}{\partial x_{i}}\right)\right)$$

The turbulence kinetic energy equation in the fluid region and turbulence kinetic energy dissipation rate equation are expressed as follows:(3)$$\rho u_{i}\frac{\partial k}{\partial x_{i}}=\frac{\partial }{\partial x_{i}}\left[\left(\mu +\frac{\mu_{t}}{\sigma_{k}}\right)\frac{\partial k}{\partial x_{i}}\right]+\frac{\mu_{t}}{2}\left(\frac{\partial u_{i}}{\partial x_{j}}+\frac{\partial u_{j}}{\partial x_{i}}\right)-\rho \varepsilon$$(4)$$\rho u_{i}\frac{\partial \varepsilon }{\partial x_{i}}=\frac{\partial }{\partial x_{i}}\left[\left(\mu +\frac{\mu_{t}}{\sigma_{k}}\right)\frac{\partial \varepsilon }{\partial x_{i}}\right]+C_{1}\frac{\mu_{t}}{2}$$
where $\;u_{i}$ and $u_{j}$ represent the mean velocity components. P and ε represent the mean pressure and dissipation rate of TKE, respectively. ρ, u, and $\mu_{t}$ are the air density, air molecular dynamic viscosity coefficient, and air turbulence dynamic viscosity, respectively.

In this model, the turbulence model is a typical turbulence model in ANSYS Fluent. Previous studies [25,26] have all used typical turbulence models to simulate heat exchanges. Therefore, the typical parameters [23] in this turbulence model are set as follows:

${Pr}_{TKE}=1$; ${Pr}_{TDR}=1.3$; ${Pr}_{wall}=0.85$; ${Pr}_{e}=0.85$; $C_{1}=1.44$; $C_{2}=1.92$; $C_{\mu }=0.09$.

The energy transport equation in the airflow is expressed as follows:(5)$$\rho_{air}C_{p,air}\frac{\partial T_{air}}{\partial_{t}}+\rho_{air}C_{p,air}u_{i}\frac{\partial T_{air}}{\partial x_{i}}=\frac{\partial }{\partial x_{i}}\left[\left(\lambda_{a}+\frac{\mu_{t}}{\sigma_{T}}\right)\frac{\partial T_{air}}{\partial x_{i}}\right]$$
where $T_{air}$, $\lambda_{a}$, and $C_{p,air}$ respectively represent the air temperature, thermal conductivity of air, and heat capacity of air. The difference in tire temperature is as follows:(6)$$∆T=T_{\max}-T_{\min}$$
where *T*_max_ and *T*_min_ are the maximum and minimum temperatures of the tire, respectively.(7)$$T_{\max}=Max\left\{T_{1},T_{2}\ldots T_{28}\right\}$$(8)$$T_{\min}=Min\left\{T_{1},T_{2}\ldots T_{28}\right\}$$

The standard deviation was calculated as follows:(9)$$s=\sqrt{\frac{1}{28-1}\sum_{i=1}^{28}\left(T_{i}-\bar{T}\right)}$$

In order to save computational resources, this paper adopts a standard turbulence model to simulate the heating process of the vulcanizing tank for retreading waste tires, following the example of some similar studies [27,28,29]. In this simulation, Patankar’s [30] SIMPLE algorithm is used to solve the pressure–velocity field. The finite volume method is employed for the discretization of the governing equations. To reduce computational resources and improve calculation speed, the momentum equation terms, turbulence kinetic energy terms, specific dissipation rate terms, and energy terms are discretized using a first-order upwind scheme, and the pressure terms are discretized using a first-order scheme. Compared with second-order discretization schemes, the first-order discretization scheme has better convergence. Although the use of this scheme helps increase discretization errors, it may lead to shorter simulation times compared to second-order schemes.

The expected criterion for solver convergence is based on the absolute residual parameter. Parameters for the continuity equation, momentum equation, kinetic energy equation, turbulence equation, and specific dissipation rate equation adopt convergence criterion 10^−3^. For the energy equation, the convergence criterion is more strictly set to 10^−5^. This means that the solver’s goal is to achieve residuals below these thresholds to ensure the convergence and accuracy of the solution to the governing equations.

**Table 2 polymers-17-00232-t002:** Thermophysical properties of 304 stainless steel and air.

Domain	Density ρ	Thermal Conductivity Coefficient k	Specific Heat Capacity c	Viscosity μ
Air calculation area	7.1587 kg/m^3^	0.02516 W/(m·K)	1.015 kJ/(kg·K)	1.78247 × 10^−5^ kg/(m·s)
Heating tube (304 SS [31])	8002 kg/m^3^	0.014 t + 14.63 W/(m·K)	0.1467 t + 495 kJ/(kg·K)	-
Retreaded tire [32,33,34]	950 kg/m^3^	−0.00048 t + 0.355 W/(m·K)	1300-0.0025 t kJ/(kg·K)	-

## 4. Grid Independence and Numerical Model Validation

To validate the numerical simulation of the vulcanization tank, this study compares the simulation results with experimental data. Figure 8 shows the comparison of the exit temperature of the heating flue gas between the experimental results and the simulation results. The mean relative error (*RE*) is expressed by Equation (10) as follows:(10)$$RE=\frac{1}{N}\sum \frac{\left|T_{exp}-T_{si}\right|}{T_{exp}}\times 100$$

In summary, the relative error between the simulation data and the experimental data are within 2.2%, further proving the accuracy of the computational method for the vulcanization tank.

To demonstrate the independence of the grid, Figure 8 shows the outlet temperatures of the heating pipe obtained using five different grid sizes. For grid sizes of 1,163,253, 1,362,248, 1,571,186, 1,811,662, and 2,140,777 cells, the relative errors between simulation and experiment are approximately 5.69%, 2.73%, 2.63%, 3.12%, 2.74%, and 1.43%, respectively. The results indicate that when the number of grids is 1,362,248 or more, the effect of grid size changes on the increase in temperature can be neglected. Therefore, to ensure more reliable results and shorter simulation times, a grid size of 1,362,248 was finally chosen for the subsequent simulations, with the simulation time unit step being consistent with the unit step of the experimental data recording set to 1 s.

In Figure 8b, a grid consisting of 1,362,248 cells was used to monitor air temperature, with different time steps of 0.02 s, 0.25 s, 0.5 s, and 1 s. The results indicate that the differences between the various time steps are minimal, with the average temperature change between the smallest and largest time steps (0.025 s and 0.5 s) being within 2.04%. Since the experimental data were recorded with a consistent time step of 1 s, we considered setting the time step to 1 s in order to ensure numerical stability while conforming to the experimental recording step size.

## 5. Results and Discussion

In this section, for the simulation of heating in the vulcanization tank under different heating powers, this paper compares the heating rate at fixed points, the temperature of the vulcanizing agent, the overall heating rate of the tire, the temperature difference analysis of the tire, as well as the temperature and velocity distribution diagrams. Subsequently, in order to improve the uniformity of temperature distribution to enhance the recycling rate of waste tires, this paper conducts a simulation analysis for the airflow organization of two types of orifice plate structures and compares the simulation results with those of the original model, proving that the orifice plates play a significant role in improving the uniformity of temperature distribution.

### 5.1. Model Validation

Figure 9a,b shows the temperature rise process of the thermometer temperature and tire temperature under different heating powers. It is worth mentioning that the label “Origin-45” refers to the original experimental simulation model with a power of 45 kW. The results indicated that there is an inverse relationship between the heating power and the time required to reach the target temperature increase. In other words, the higher the heating power, the shorter the time needed for both the thermometer temperature and tire temperature needed to reach the target. The time required for the tire temperature to reach 80 °C at heating powers of 90 kW, 75 kW, 60 kW, and 45 kW is 7330 s, 8760 s, 10,910 s, and 14,490 s, respectively.

The reason for this phenomenon is that the higher the power of the heater, the higher the temperature of the heating element. Due to the increased temperature difference between the heated air and the dry-burn tube, the amount of heat absorbed by the air per unit of time increases, and thus, the thermometer and the tire absorb more heat per unit of time, leading to a faster rate of increase in temperature. Combining Figure 9a,b shows that when the heating power was 90 kW, the thermometer’s temperature had risen to 100 °C but heating had yet to stopped; instead, it continued increasing for a while before stopping because the tire’s temperature had yet to reach the target temperature (80 °C).

Similarly, Figure 9c presents the temperature profile of the vulcanizing agent during the heating process of the vulcanizing tank. The results demonstrate that increasing the heating power helps the vulcanizing agent reach the vulcanization temperature sooner, thereby shortening the retreading time. Specifically, the time required to reach the vulcanization temperature with a heating power of 90 kW is almost 40% of that required with a heating power of 45 kW.

Figure 9d shows the temperature difference diagram of the vulcanizing agent during the heating process. It can be observed from Figure 9d that in the early stage of heating, an increase in heating power at the same time point leads to an increase in the maximum temperature difference, which may be due to the existence of a heat transfer blind zone with air flow inside the tire and excessive heat concentration. However, it can be noted that in the later stage of heating, at the same time point, the higher the heating power, the smaller the temperature difference. This is because the higher the heating power, the easier it is for the vulcanization tank to reach the conditions required to stop heating, and then the tire, under the combined effects of forced convection and natural convection, carries away the heat from the higher temperature parts of the tire, causing the temperature difference to drop rapidly.

Figure 10 and Figure 11 show the xy slice temperature distribution and velocity distribution during the heating process of the vulcanization tank with a heating power of 60 kW. It can be observed from the figures that at times of 3000 s, 6000 s, and 9000 s, both the highest and lowest temperatures increased significantly. The temperature difference across the slice increased, and the general temperature distribution of the tire was from the bottom to the top and from right to left. The temperature then gradually decreased, with the right-bottom tread being higher in temperature and the left-top tread being lower in temperature. By analyzing the velocity distribution, it can be found that the right side of the vulcanization tank under the tire receives significant concentrated heating, while the upper side of the left tire lacks heating. In addition, the internal air temperature on the left side of the tire is significantly lower than that in other areas, indicating an uneven distribution of air temperature.

For a time period of 12,000 s, the maximum temperature and the section temperature decreased significantly. This is because the heating stop condition was reached at *t* = 10,900 s, but the fan continued to work, and forced convection combined with natural convection promoted heat transfer, reducing the temperature difference. It is worth noting that at that time, the minimum temperature on the left tire was 65 °C, which had yet to reach the vulcanization temperature. The temperature on the right side reached 89 °C.

In summary, the increase in heating power helps reduce the heating time required for the vulcanizing agent, the vulcanization tank thermometer temperature, and the tire temperature needed to reach the target. However, this also increases the temperature difference on the tread, leading to concentrated heating on the lower part of the right tire, which may cause over-vulcanization or under-vulcanization in some areas of the tread, thus affecting the quality of the retreaded tires.

### 5.2. Analysis of the Optimization Results for Airflow Organization

In order to adjust the heat accumulation caused by airflow and ensure the uniform heating of waste tires, this paper proposed two orifice plate structures to adjust the airflow organization flow, as shown in Figure 4 and Figure 5. Figure 12 shows the temperature heating curves of two types of vulcanizers with and without orifice plates. “Origin” refers to the untreated vulcanizer, while “Flat” and “Frustum” are the vulcanizers with flat orifice plates and frustum cone orifice plates.

From the graph, we can observe that the thermometer temperature of the vulcanization tank with the orifice plate treatment first rises, then levels off, and finally decreases. This is because the thermometer had reached the target temperature, but the overall temperature of the tire had yet to reach the vulcanization temperature, and the heating temperature had to be maintained at 100 °C so that the tire temperature reached 80 °C before heating stopped, causing the thermometer temperature to decrease.

Compared to the untreated vulcanization tank, the vulcanization tank with orifice plates reaches the target temperature earlier, and the time required to heat to the target temperature decreases as the heating temperature increases. This is because the presence of the orifice plate slows down the flow velocity of the fluid near the thermometer, leading to the rapid accumulation of heat and a quick increase in temperature, ultimately reaching the target temperature. Additionally, in the vulcanization tank with orifice plates, the time required to reach the target temperature is similar for frustrum cone orifice plates and flat orifice plates, but the frustrum cone orifice plate reaches the point of stopping heating sooner than the flat orifice plate, and the distance increases with the rise in heating power. This means that when the heating power is 45 kW, 60 kW, 75 kW, and 90 kW, the presence of the orifice plate can approximately save at least 90 kWh, 68 kWh, 50 kWh, and 36 kWh, respectively.

Figure 13 shows the heating process diagrams of the tires in three different types of vulcanization tanks. The results indicate that when the heating power is 45 kW, the time required for the waste tires in the vulcanization tank with orifice plates to reach the vulcanization temperature is approximately 7800 s, while for the untreated vulcanization tank waste tires, it takes 14,500 s, which is directly reduced by 46.2%. The time difference between the two decreases with the increase in heating power. When the heating power is 90 kW, the time required for the waste tires in the vulcanization tank with orifice plates to reach the vulcanization temperature is about 5900 s, and for the untreated vulcanization tank waste tires, it takes 7300 s, which is reduced by 19.2%. The results show that the orifice plate adjusts the flow of the heating air, making the heating air more concentrated towards the tires, thus making it easier for the tires to reach the target temperature. In the comparison of the time required for the two types of orifice plate tires to reach the vulcanization temperature, the frustum cone orifice plate has a slight advantage over the flat orifice plate, with a time difference ranging from 200 to 480 s.

Figure 14 shows the temperature heating history of the vulcanizing agent under different power levels. The process of retreading old tires actually involves heating the vulcanizing agent to the vulcanization temperature to bond the old tire with a new tread, creating a new tire. Under different heating powers, the effects of the frustrum cone orifice plates and flat orifice plates on increasing the temperature of the vulcanizing agent are almost the same, which may be due to the similarity in their structures. The time required for the vulcanizing agent with orifice plates to reach the vulcanization temperature is approximately 7300 s, while the time required for the untreated vulcanizing agent to reach the vulcanization temperature is about 14,200 s, which means that at a heating power of 45 kW, the presence of the orifice plates can help reduce the time required to reach vulcanization temperature by 48.6%. Similarly, this time difference decreases with the increase in heating power, reaching a reduction of 20.7% at a heating power of 45 kW. The results indicate that the presence of the orifice plates regulates the flow of the heating air, making it more concentrated towards the vulcanizing agent, thus facilitating the vulcanizing agent to reach the vulcanization temperature more easily.

A large number of studies [35,36,37,38] have shown that the retreading results of tires are related to the temperature of the vulcanizing agent, and an excessively large temperature difference can easily lead to the over-vulcanization reaction of the vulcanizing agent in some positions, which affects the quality of retreaded tires. Figure 15 shows the heating process of the vulcanizing agent temperature difference at different powers. The results show that the difference in the temperature of the vulcanizing agent treated with the orifice plate can greatly reduce the maximum temperature difference. At the same time, with an increase in heating power, the difference between the temperature difference of the vulcanizing agent treated with the orifice plate and the difference in the temperature of the vulcanizing agent without an orifice plate will increase. This means that as the heating power increases, the temperature regulation of the orifice plate becomes more obvious. In the comparison of orifice vulcanization tanks, the temperature regulation of the frustrum cone orifice plate is superior to that of the flat orifice plate, resulting in the better quality of the retread tires.

Figure 16 shows the heating process diagram of the standard deviation of the vulcanizing agent temperature under different powers. When the heating power is 45 kW, the standard deviation of the flat plate orifice plate and the frustum cone orifice plate is 7.01, which is 7.2 lower than that of the untreated vulcanization tank. When the heating power is 90 kW, the standard deviation of the flat plate orifice plate and the frustum cone orifice plate is 11.1, which is 11.75 lower than that of the untreated vulcanization tank. The results indicate that the presence of the orifice plates can greatly improve the uneven temperature distribution of the vulcanizing agent, thereby enhancing the quality of retreaded tires, and the regulation effect becomes more pronounced with the increase in heating power.

Figure 17 and Figure 18 show the cross-sectional xy slice temperature distribution of a vulcanizer with orifice plates during the heating process at a heating power of 60 kW. As can be seen from the diagram, when the time is 3000 s, there is still a slight trend of the temperature inside the vulcanizing tank being lower than outside. However, at 6000 s, this phenomenon improves, and the temperature difference between the inside and outside of the retreaded tire is significantly reduced. By 9000 s, this phenomenon is further reduced. At 12,000 s, the temperatures inside and outside of the retreaded tire are essentially consistent. The results indicate that the presence of the orifice plate and the frustrum cone orifice plate can effectively improve the uneven temperature distribution issue in the retreaded tire vulcanization process, thereby enhancing the quality of the retreaded tires.

Additionally, Figure 19 and Figure 20 show the velocity distribution diagrams of the cross-sections of the xy slice during the heating process of the vulcanization tank with frustum cone orifice plates and flat orifice plates at a heating power of 60 kW. The results indicate that the orifice plates can mitigate the issue of excessive heat concentration.

Although the improvement effect of the frustum cone orifice plate is better than that of the flat orifice plate, the area of the frustum cone orifice plate is 76% higher than that of the flat orifice plate, which means that the manufacturing cost of the frustum cone orifice plate is 76% higher than that of the flat orifice plate. Moreover, the shape of the frustum cone orifice plate is more complex than that of the flat orifice plate, which implies higher costs.

The results indicated that the presence of the orifice plate prevents heat accumulation and addresses the issue of excessive temperature difference during the heating of used tires, which can cause over-vulcanization or under-vulcanization, thereby enhancing the quality of the retreaded tires. In the future, the structure of the orifice plate can be optimized to concentrate heat more effectively on the vulcanizing agent, reducing the heating time and retreading duration of the tires, ultimately achieving energy savings and efficiency improvements.

## 6. Conclusions

The main conclusions of this study are as follows:(1)During the heating process of a vulcanization tank without orifice plate, an increase in heating power leads to a decrease in the time required for the thermometer temperature and the tire to reach the target temperature, as well as for the vulcanizing agent to reach the vulcanization temperature. This helps improve the renovation rate of retreaded tires. However, the increase in heating power can also lead to the accumulation of heat, resulting in an increase in the temperature difference within the vulcanizing agent and an enlargement of the standard deviation at the monitoring points. This can cause the over-vulcanization or under-vulcanization of the retreaded tire, ultimately affecting its quality.(2)During the heating process, placing orifice plates in the vulcanization tank can significantly improve the uneven temperature distribution issue that occurs during the original vulcanization tank heating process, greatly reducing the difference in the vulcanizing agent temperature and the standard deviation of the monitoring points for the vulcanizing agent. This mitigates the over-vulcanization or under-vulcanization of retreaded tires, enhancing the quality of retreaded tires. At the same time, the presence of the orifice plate also reduces the time required for the thermometer temperature to reach the target temperature of the tire heating and the vulcanizing agent heating to the vulcanization temperature, thereby reducing the heating time and ultimately achieving energy savings. When the heating power is 45∼90 kW, the presence of the orifice plate can approximately save at least 90~36 kW*h, respectively.(3)In the comparison of orifice plates, the performance of the frustum cone orifice plate is shown to be better than that of the flat orifice plate. However, the area of the frustum cone orifice plate is 76% higher than that of the flat orifice plate, and it is more complex to manufacture, increasing costs. Therefore, considering all factors, this paper recommends using the flat orifice plate for improving the uneven temperature distribution issue during the vulcanization tank heating process.

## Figures and Tables

**Figure 1 polymers-17-00232-f001:**
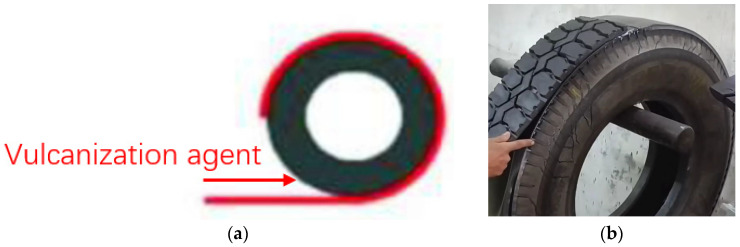
Operation diagram of bonding rubber sheets from waste tire vulcanizing agents. (**a**) Position of vulcanizing agent; (**b**) tire retread operation diagram.

**Figure 2 polymers-17-00232-f002:**
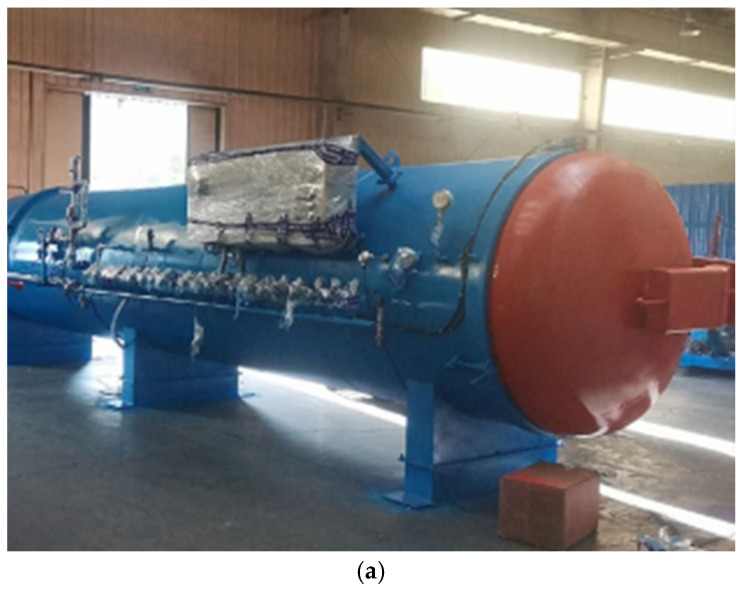
Experimental equipment. Vulcanization tank: (**a**) physical drawing; (**b**) top view and front view; (**c**) side view.

**Figure 3 polymers-17-00232-f003:**
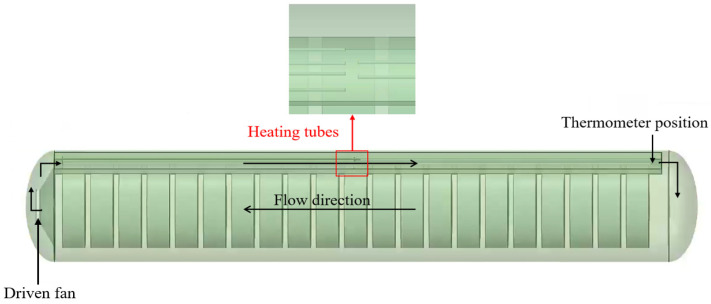
Flow direction and thermometer position.

**Figure 4 polymers-17-00232-f004:**
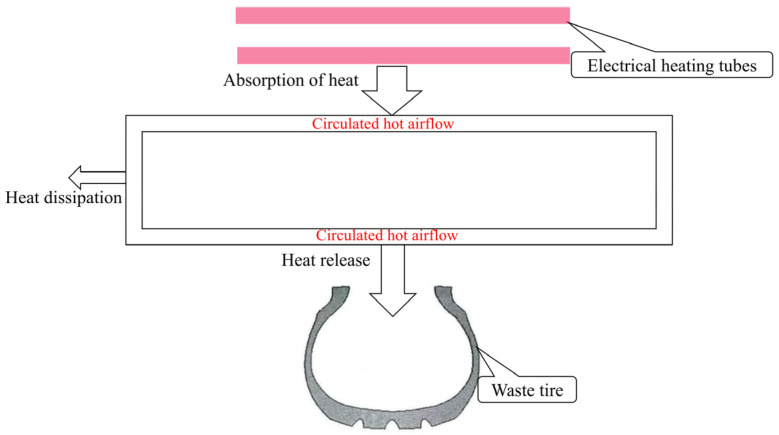
Schematic diagram of energy transfer.

**Figure 5 polymers-17-00232-f005:**
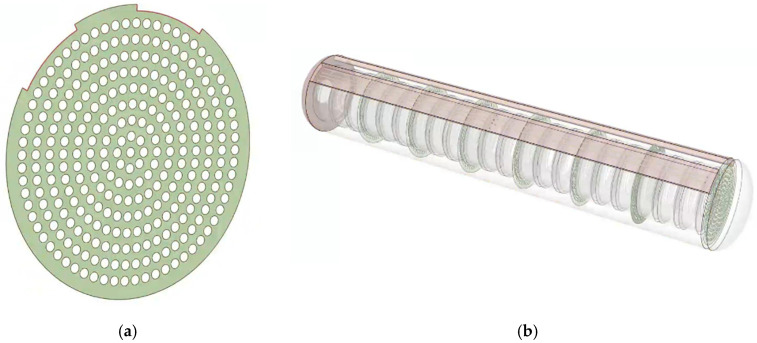
Schematic diagram of the plain orifice plate: (**a**) flat orifice plate; (**b**) vulcanization tank with flat orifice plates.

**Figure 6 polymers-17-00232-f006:**
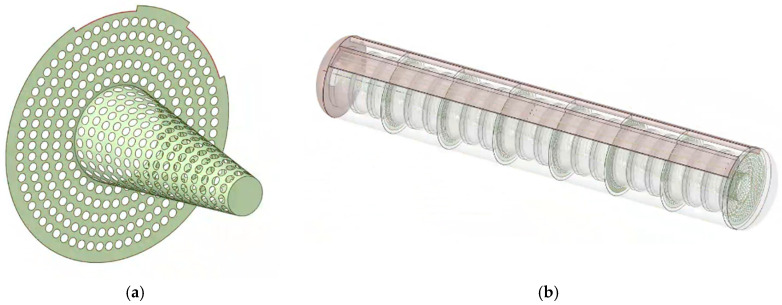
Schematic diagram of the plain orifice plate: (**a**) frustum cone orifice plate; (**b**) vulcanization tank with frustum cone orifice plates.

**Figure 7 polymers-17-00232-f007:**
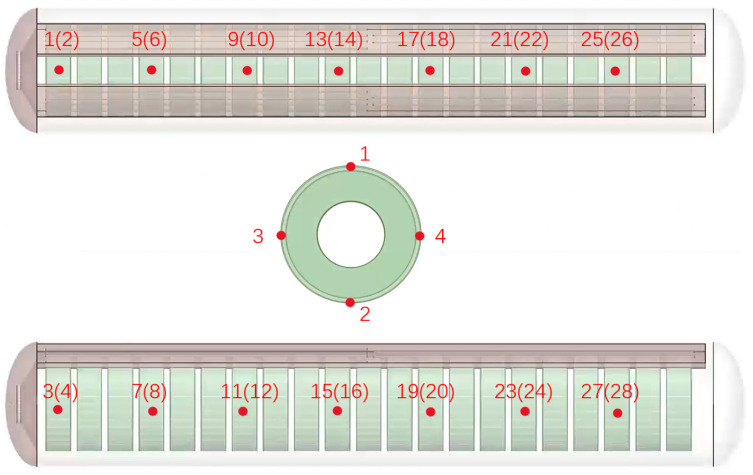
Location diagram of the temperature monitoring positions for the vulcanizing agent.

**Figure 8 polymers-17-00232-f008:**
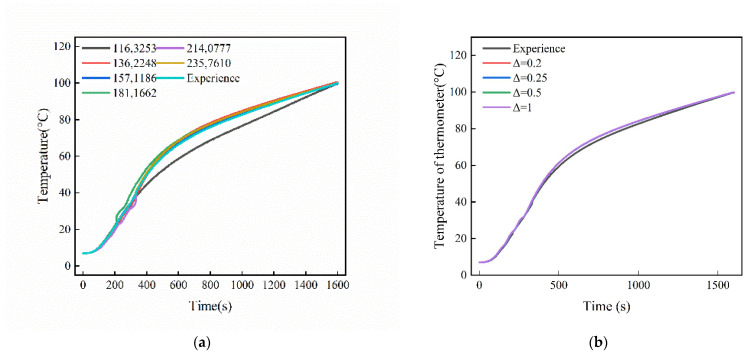
Grid and time independence verification: (**a**) grid independence verification; (**b**) time independence verification.

**Figure 9 polymers-17-00232-f009:**
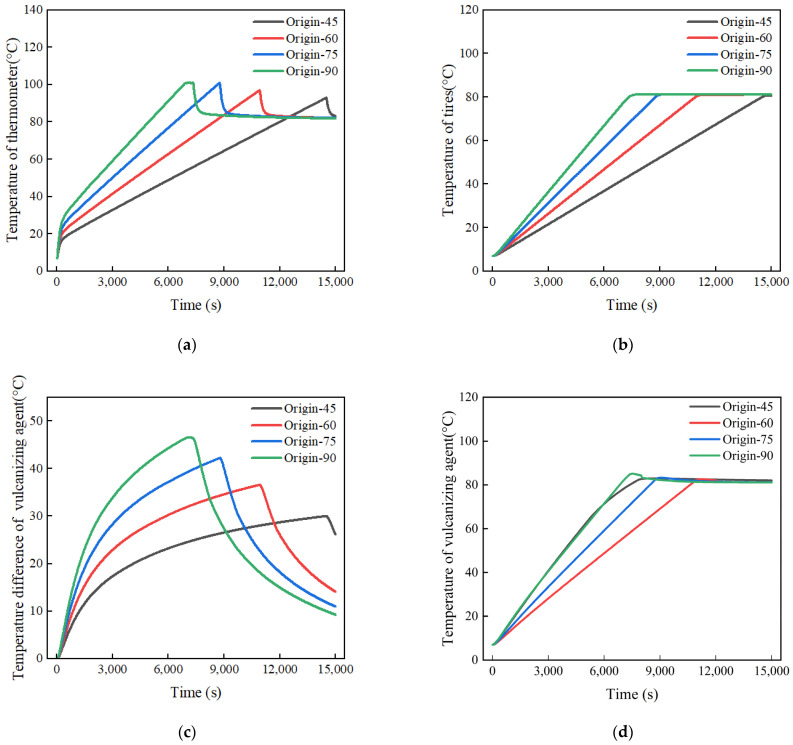
Temperature history: (**a**) temperature history of the thermometer; (**b**) temperature history of the tire; (**c**) temperature difference history of the vulcanizing agent; (**d**) temperature history of the vulcanizing agent.

**Figure 10 polymers-17-00232-f010:**
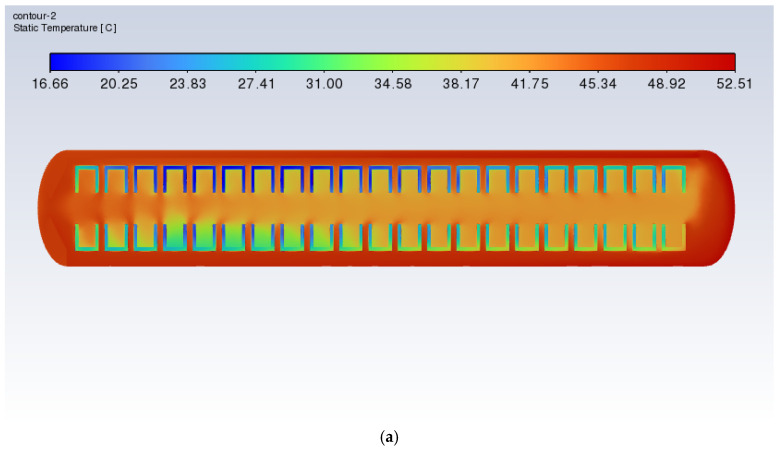
Cross-sectional xy slice temperature distribution of the vulcanization tank (60 kW): (**a**) time = 3000 s; (**b**) time = 6000 s; (**c**) time = 9000 s; (**d**) time = 12,000 s.

**Figure 11 polymers-17-00232-f011:**
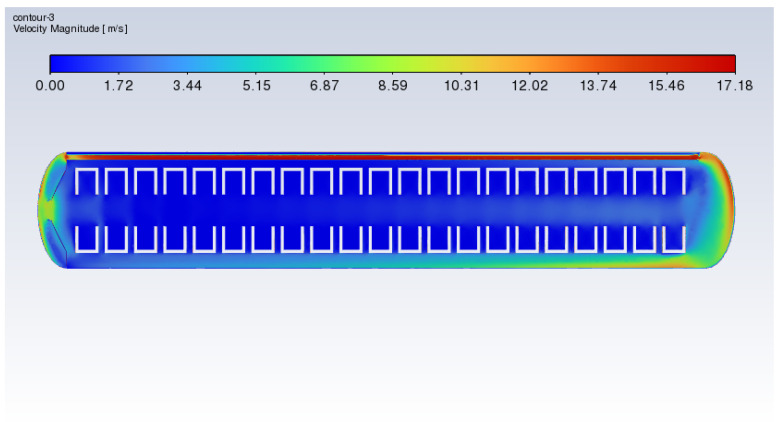
Cross-sectional xy slice velocity distribution of the vulcanization tank (60 kW, 3000–12,000 s).

**Figure 12 polymers-17-00232-f012:**
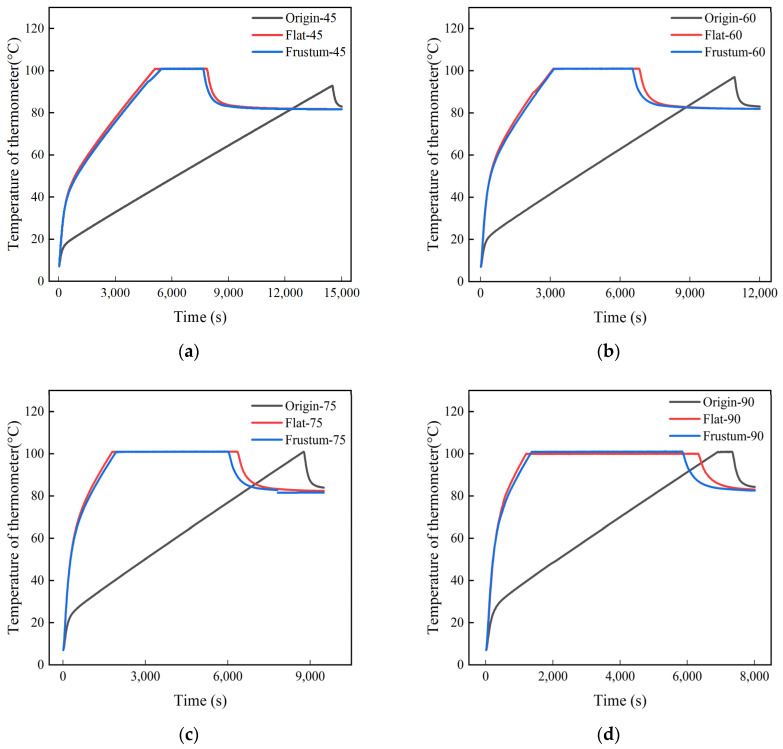
Temperature history diagrams of thermometers under different power levels: (**a**) 45 kW; (**b**) 60 kW; (**c**) 75 kW; (**d**) 90 kW.

**Figure 13 polymers-17-00232-f013:**
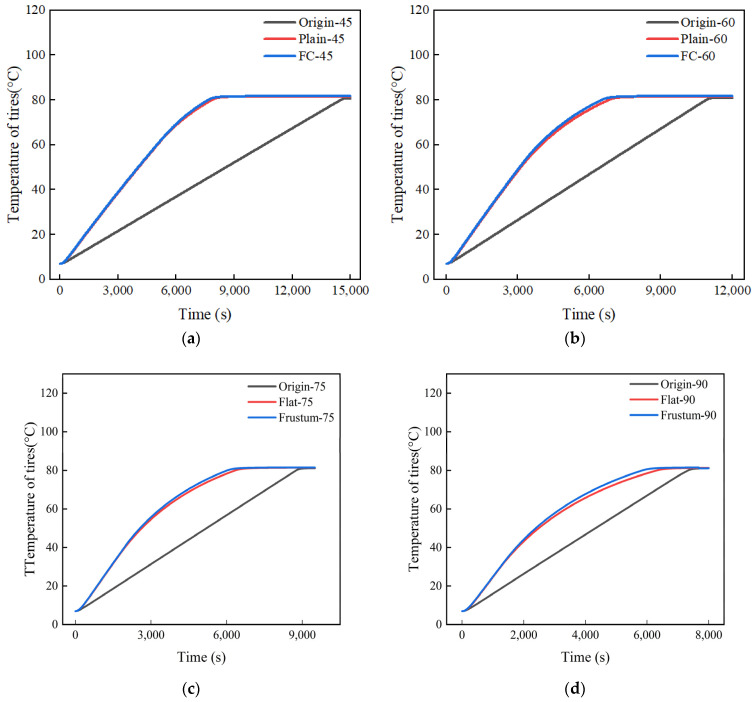
Temperature history diagrams of tire under different power levels: (**a**) 45 kW; (**b**) 60 kW; (**c**) 75 kW; (**d**) 90 kW.

**Figure 14 polymers-17-00232-f014:**
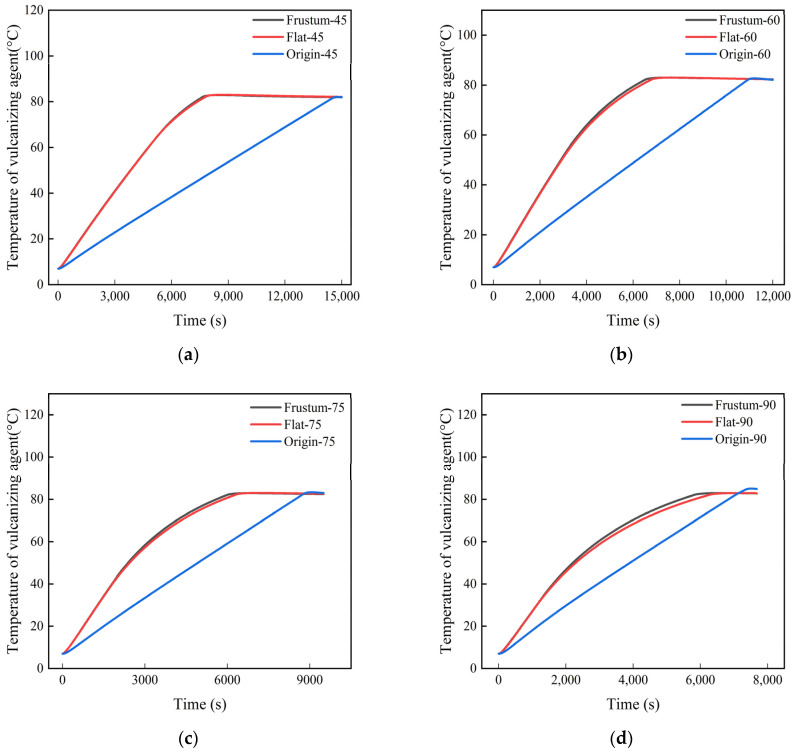
Temperature history diagrams of the vulcanizing agent under different power levels: (**a**) 45 kW; (**b**) 60 kW; (**c**) 75 kW; (**d**) 90 kW.

**Figure 15 polymers-17-00232-f015:**
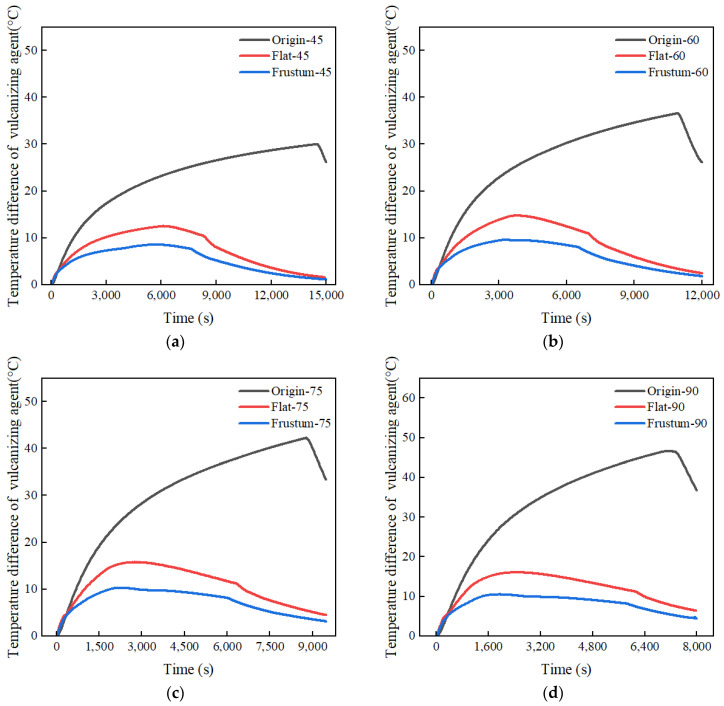
Temperature difference heating process diagram of the vulcanizing agent under different powers: (**a**) 45 kW; (**b**) 60 kW; (**c**) 75 kW; (**d**) 90 kW.

**Figure 16 polymers-17-00232-f016:**
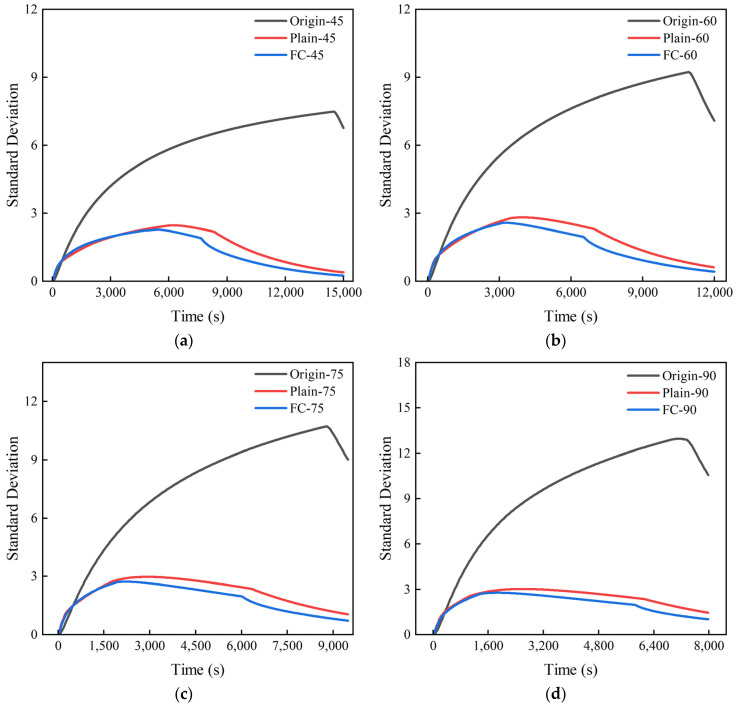
Standard deviation heating process diagram of the vulcanizing agent under different powers: (**a**) 45 kW; (**b**) 60 kW; (**c**) 75 kW; (**d**) 90 kW.

**Figure 17 polymers-17-00232-f017:**
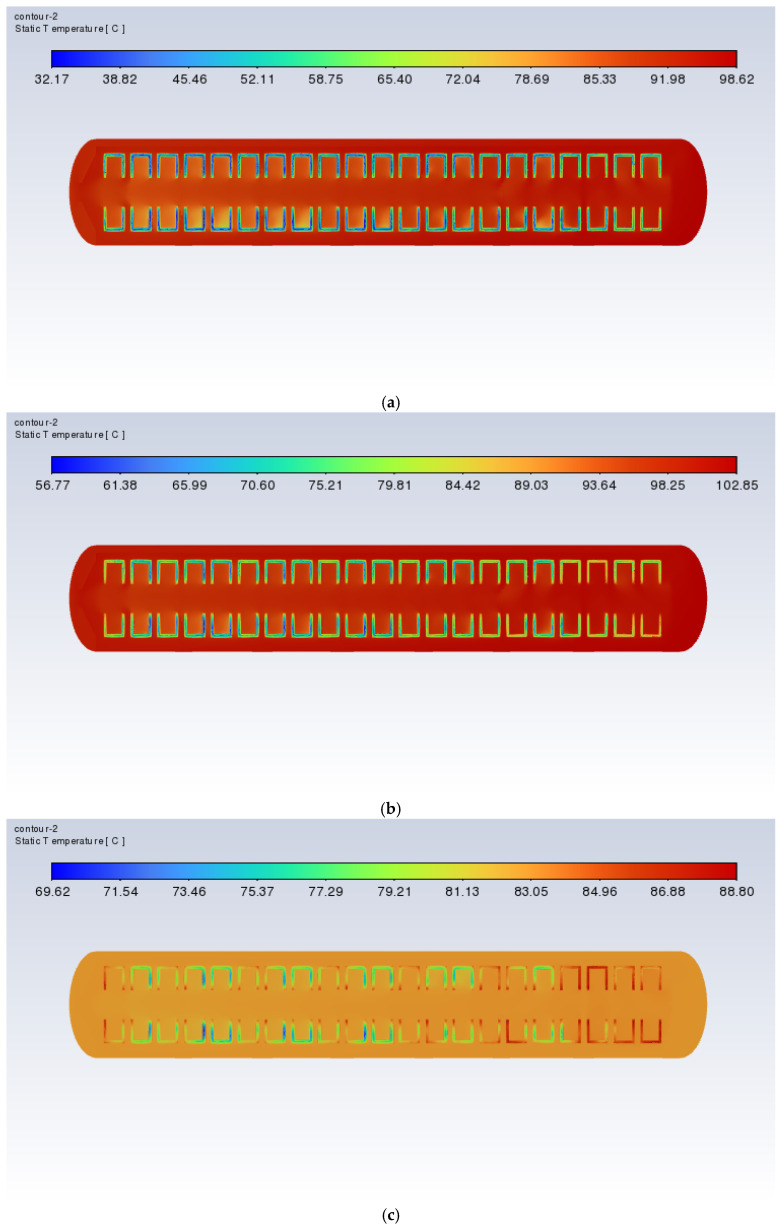
Cross-sectional xy slice temperature distribution of the vulcanization tank with flat plates (60 kW): (**a**) time = 3000 s; (**b**) time = 6000 s; (**c**) time = 9000 s; (**d**) time = 12,000 s.

**Figure 18 polymers-17-00232-f018:**
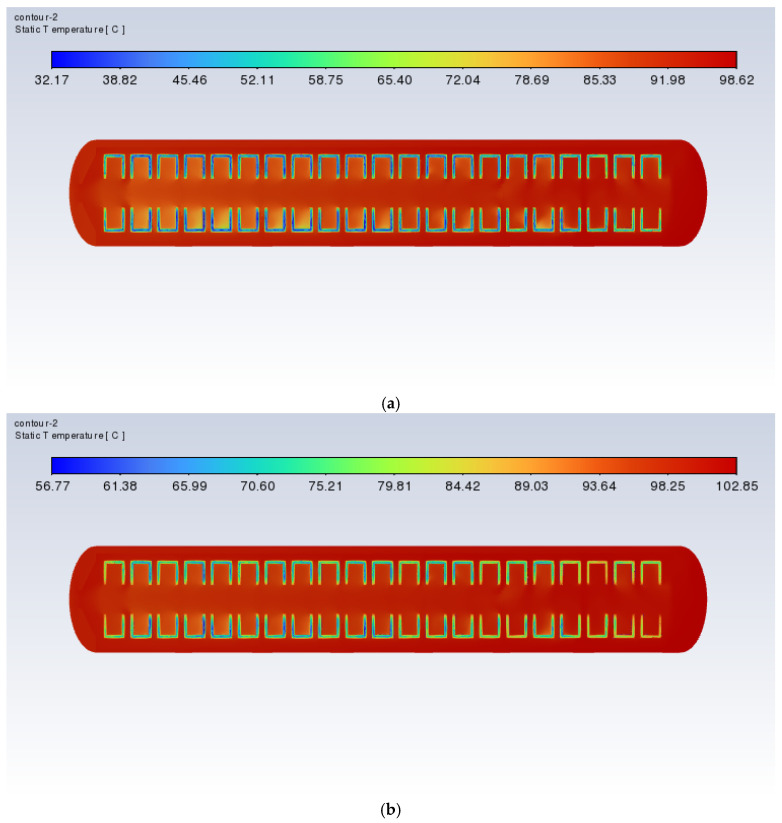
Cross-sectional xy slice temperature distribution of the vulcanization tank with frustum cone plates (60 kW): (**a**) time = 3000 s; (**b**) time = 6000 s; (**c**) time = 9000 s; (**d**) time = 12,000 s.

**Figure 19 polymers-17-00232-f019:**
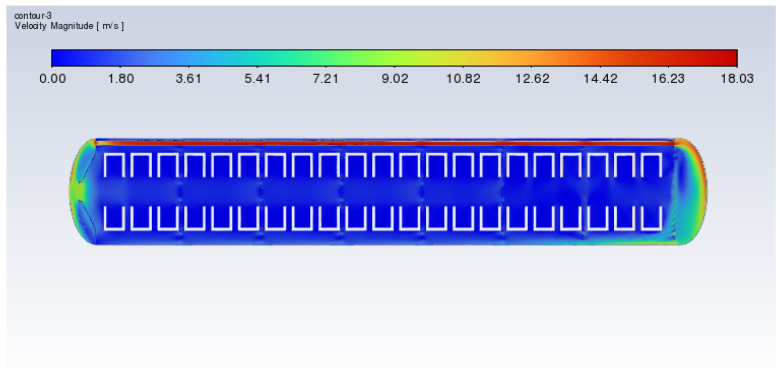
Cross-sectional xy slice velocity distribution of the vulcanization tank with flat plates (60 kW).

**Figure 20 polymers-17-00232-f020:**
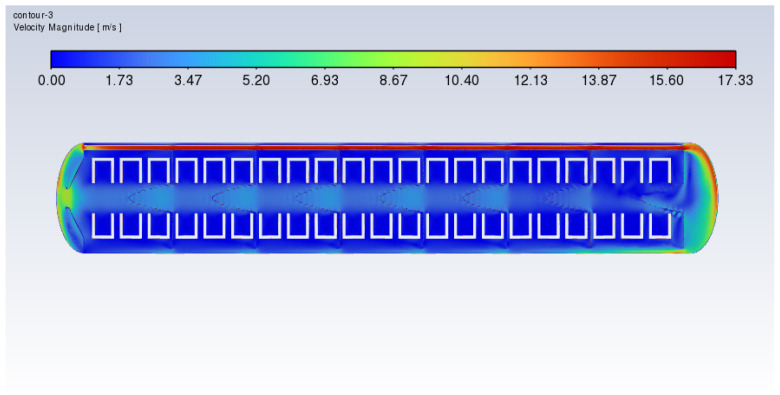
Cross-sectional xy slice temperature distribution of the vulcanization tank with frustrum cone orifice plates (60 kW).

**Table 1 polymers-17-00232-t001:** Parameters of the vulcanization tank used in the experiment.

Parameters	Values
L_1_	8500 mm
L_2_	8400 mm
L_3_	4063 mm
L_4_	400 mm
L_5_	200 mm
D_1_	1600 mm
D_2_	487.2 mm
R_1_	800 mm
R_2_	700 mm
r_1_	30°
r_2_	33.4°

## Data Availability

Data are contained within the article.

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
