# Peer review of "Numerical Simulation of Airflow Organization in Vulcanization Tanks for Waste Tires"

_polymers, 2025, doi:10.3390/polym17020232_

Round 1
Reviewer 1 Report
Comments and Suggestions for Authors
This paper is well written and comprehensive. The results presented are of interest for rubber industry and production planing. I have marked minor suggestions in pdf file attached.

Author Response
Comments 1:Change 'of' to 'and'.
Response 1:Agree. We had completed it on line 26.
Comments 2:Please provide reference.
Response 2:Agree. We had completed it on line 33.
Comments 3:Superscript.
Response 3:Agree. We had completed it on line 60,301, 302 and 249.
Comments 4:Provide the meaning of abbreviation.
Response 4:Agree. We had completed it on line 52.
Comments 5:Change 'Experimental Equipment' to 'Materials and Methods'.
Response 5:Agree. We had completed it on line 143.
Comments 6:Space before bracket.
Response 6:Agree. We had completed it on line 170,202 and 425.
Comments 7:The temperature of the.
Response 7:Agree. We had completed it on line 170.
Comments 8:Which software was used?
Response 8:Agree. We had mentioned it on line 134,191,224.
Comments 9:Change 'the Standard deviation formula' to 'Standard deviation was calculated'.
Response 9:Agree. We had completed it on line 233.
Comments 10:Please use adequate symbol for multiplication.
Response 10:Agree. We had correct it on table 2.
Comments 11:This is already mentioned above.
Response 11:Agree. We had removed it on line 256.
Comments 12:Italic.
Response 12:Agree. We had correct it on line 258,353.
Comments 12:This can be removed.
Response 12:Agree. We had removed it on line 503.
Comments 13:Please rewrite this sentence.
Response 13:Agree. We had rewrite it on line 303.
Comments 14:Change 'Result' to 'Result and Discussion'.
Response 14:Agree. We had rewrite it on line 280.
Comments 15:Please check symbols. They are different in above equations.
Response 15:Agree. We had correct it on line 220,221 and 222.
Comments 16:Change 'When the time is ' to 'For the time of '.
Response 16:Agree. We had correct it on line 351.
Comments 17:Change 'quality ' to 'quality of '.
Response 17:Agree. We had correct it on line 437.
Comments 18:Removed ' thermometer'.
Response 18:Agree. We had correct it on line 373.
Comments 19:Please correct units.
Response 19:Agree. We had correct it on line 388,389.
Comments 20:Removed '6. Discussion '.
Response 20:Agree. We had removed it.

Reviewer 2 Report
Comments and Suggestions for Authors
1 - The idea is very interesting, but methods are not well explained and introduced,
2 - the software that you used FEM software I believe is not introduced,
3 - and really the process in the cilindric structure is not well defined
4- and vulcanization agent neither defined
5- and its position is not well explained, please introduce changes in this areas,
6- introduce a process diagram specifiinf very well this,
7 - neither there is no information If this research is for a company or project, 8- and I believe that for this is important to explain and clarify a energy balance about the retraitment process and energy diagram about the process of retraitment described by you
Describe the benefit of the 2 retraitment processes described by you (high temperature and low temperature)
Author Response
Comments 1: The idea is very interesting, but methods are not well explained and introduced.
Response 1: Thank you for your suggestion. We have completed the modifications.
Comments 2: The software that you used FEM software I believe is not introduced.
Response 2: We agree with this comment. Therefore, we made modifications and introductions on lines 134, 191, and 224.
Comments 3: And really the process in the cilindric structure is not well defined.
Response 3: We agree with this comment. Therefore, we made modifications in Figure 3 and introduced it on line 165.
Comments 4: Vulcanization agent neither defined.
Response 4: We agree with this comment. Therefore, we made an introduction from line 80 to 85.
Comments 5: And its position is not well explained, please introduce changes in this areas.
Response 5: We agree with this comment. Therefore, we made an introduction on line 79.
Comments 6: Introduce a process diagram specifiinf very well this,.
Response 6: We agree with this comment. Therefore, we made a modification in Figure 1.
Comments 7:Neither there is no information If this research is for a company or project
Response 7: In facet. The author, Baolin Wang, is the CEO of the company, which can be seen on the website, https://aiqicha.baidu.com/company_detail_20492703234228. And Xiaowen Luan and Hui Li is the employee of the company.
Comments 8: and I believe that for this is important to explain and clarify a energy balance about the retraitment process and energy diagram about the process of retraitment described by you.
Response 8: We agree with this comment. Therefore, we added a schematic diagram of energy transfer in Figure 4.
Comments 9: Describe the benefit of the 2 retraitment processes described by you.
Response 8: We agree with this comment. The article has already mentioned the benefits of cold vulcanization. In addtion, we also introduced the benefits of hot vulcanization on lines 111 to 113.
Round 2
Reviewer 2 Report
Comments and Suggestions for Authors
Thank you for your answers
Comments on the Quality of English LanguageEnglish seems in good level